# Preoperative Localization for Primary Hyperparathyroidism: A Clinical Review

**DOI:** 10.3390/biomedicines9040390

**Published:** 2021-04-06

**Authors:** Donovan Tay, Jeeban P. Das, Randy Yeh

**Affiliations:** 1Department of Medicine, Sengkang General Hospital, 110 Sengkang E Way, Singapore 544886, Singapore; donovan.tay.y.k@singhealth.com.sg; 2Department of Radiology, Molecular Imaging and Therapy Service, Memorial Sloan Kettering Cancer Center, 1275 York Avenue, New York, NY 10065, USA; dasj@mskcc.org

**Keywords:** preoperative localization, primary hyperparathyroidism, ultrasound, sestamibi, 4-dimension computed tomography, 4DCT, positron emission tomography, PET, magnetic resonance imaging, MRI, parathyroid adenoma, preoperative imaging

## Abstract

With increasing use of minimally invasive parathyroidectomy (PTx) over traditional bilateral neck exploration in patients with primary hyperparathyroidism (PHPT), accurate preoperative localization has become more important to enable a successful surgical outcome. Traditional imaging techniques such as ultrasound (US) and sestamibi scintigraphy (MIBI) and newer techniques such as parathyroid four-dimension computed tomography (4D-CT), positron emission tomography (PET), and magnetic resonance imaging (MRI) are available for the clinician to detect the diseased gland(s) in the preoperative workup. Invasive parathyroid venous sampling may be useful in certain circumstances such as persistent or recurrent PHPT. We review the diagnostic performance of these imaging modalities in preoperative localization and discuss the advantages and weaknesses of these techniques. US and MIBI are established techniques commonly utilized as first-line modalities. 4D-CT has excellent diagnostic performance and is increasingly performed in first-line setting and as an adjunct to US and MIBI. PET and MRI are emerging adjunct modalities when localization has been equivocal or failed. Since no evidence-based guidelines are yet available for the optimal imaging strategy, clinicians should be familiar with the range and advancement of these techniques. Choice of imaging modality should be individualized to the patient with consideration for efficacy, expertise, and availability of such techniques in clinical practice.

## 1. Introduction

### 1.1. Clinical Relevance of Primary Hyperparathyroidism

Primary hyperparathyroidism (PHPT) is an endocrine disorder resulting from excess secretion of parathyroid hormone (PTH), from one or more of the parathyroid glands. It accounts for most cases of hypercalcemia and has a prevalence of ~3 in 1000 within the general population. It is most commonly seen in postmenopausal women [1].

PHPT can result in multisystem complications, most frequently involving bones and kidneys. Dr. Fuller Albright famously and succinctly characterized PHPT as a disease of “bones and stones”. While osteitis fibrosa cystica is the “classic” osseous manifestation of PHPT resulting in bone pain, fractures, and deformities, renal complications typically result in the development of nephrolithiasis and/or nephrocalcinosis. With the advent of increased biochemical testing, the clinical presentation for PHPT has evolved from a challenging multisystem symptomatic disease process to a primarily asymptomatic chronic illness. Biochemical diagnosis is established based on laboratory findings of hypercalcemia with elevated or inappropriately normal levels of PTH. Despite the lack of clinical symptoms, the detrimental effects of PHPT on the skeleton and kidneys, remain pervasive. For example, fracture risk is increased by a factor of 1.5 in asymptomatic individuals with PHPT versus the general population [2], with vertebral fractures found in approximately 20% of asymptomatic PHPT patients [3,4]. Similarly, osteoporosis is present in 66% of asymptomatic patients with PHPT [4]. On densitometric assessment, there is preferential loss of bone mineral density in the distal radius with relative preservation of the lumbar spine bone mineral density [5] In addition, asymptomatic nephrolithiasis is found in 20% of PHPT patients [4,6]. If untreated, asymptomatic PHPT may lead to an acceleration of decreased bone mineral density over time resulting in a higher fracture risk and enlarging or migrating renal calculi may eventually cause symptoms, lead to obstruction and/or renal impairment [7].

Definitive surgical management with parathyroidectomy (PTx) ameliorates the detrimental effects of PHPT on the skeleton and kidneys. While the vast majority of PHPT cases occur due to a solitary sporadic parathyroid adenoma (single-gland disease [SGD]), a smaller proportion are as a result of a double adenoma or hyperplasia (multigland disease [MGD]) [8,9].

### 1.2. Clinically Relevant Embryology of the Parathyroid Glands

Localizing the abnormal parathyroid gland (or glands) is critical to successful PTx and knowledge of the typical and atypical anatomical locations of the parathyroid glands is important in successful localization. Generally, there are four normal glands in place, although supernumerary glands can occur infrequently [10], up to 7% in one meta-analysis [11]. Embryologically, the inferior glands arise from the third brachial pouch. During embryological development, they descend along with the thymus to their final typical eventual position around the lower pole of the thyroid gland, just anterior to the vertical plane of the recurrent laryngeal nerve (RLN). As a result of this longer descent (as compared with the superior glands), they are typically more variable in location and are more likely to be ectopic [12]. Examples of ectopic inferior locations include within the thyrothymic ligament, in the thymus itself or otherwise within the mediastinum. Rarely, ectopic inferior glands may be seen close to the carotid bifurcation [13].

Paired superior glands arise from the fourth brachial pouch and are relatively more consistent in its anatomic location, due to the shorter descent and migration with the thyroid gland. Typically, they are located behind the mid-to- upper pole of the thyroid gland, 1cm above the junction between the inferior thyroid artery and the RLN at the level of the cricoid cartilage gland, posterior to the plane of the RLN [10]. It is important to note that larger and heavier superior parathyroid adenoma can displace inferiorly over time due to gravity and at times end up located more inferiorly than the inferior parathyroid glands. However, while the superior–inferior relationship is disrupted, the anterior–posterior relationship of the RLN is consistently maintained, with inferior glands lying anterior to the RLN and superior glands lying posterior to the RLN. Ectopic locations of the superior glands include retropharyngeal or intrathyroidal location, the latter being accounted for by the fact that the thyroid C cells also originate from the fourth brachial pouch during embryogenesis [12,14]. A recent meta- analysis demonstrated that up to 16% of glands are found in ectopic locations: ~12% in the neck and ~4% in the mediastinum. Of ectopic glands in the neck, the majority are found in retroesophageal/paraesophageal space (31.4%), intra-thyroidal (20.3%), carotid sheath (17.7%), thyrothymic ligament (17%), tracheoesophageal groove (5.1%), and 8.4% in other locations (thyroid cartilage, retropharyngeal space, and adjacent to hyoid bone). The majority of ectopic glands found in the mediastinum are in thymus [11]. The weight of normal glands typically ranges from 35 to 40 mg based on prior cadaveric series [14], which may be extrapolated as a cutoff for imaging studies [15].

### 1.3. Role of Imaging in PHPT

Over the past two decades, unilateral minimally invasive parathyroidectomy (MIP) has largely replaced traditional bilateral neck exploration (BNE) as the surgery of choice for PTx, with the rationale that most patients (>80%) will have SGD, precluding the need for more extensive surgical exploration [16,17]. Compared to BNE, MIP decreases complication rates (i.e., vocal cord paralysis), operating and anesthesia time, while also improving cosmetic results, and providing similar surgical cure rates. This shift in surgical practice from BNE to MIP has emphasized the importance of preoperative imaging, as accurate localization is critical to the success of MIP. For the surgeon, imaging plays a dual role by localizing abnormal gland(s) and stratifying patients who qualify for MIP from those who would be better served with BNE [17]. Figure 1 demonstrates a flowchart algorithm for how imaging is used to guide surgical management.

MIP is performed in patients with SGD (solitary adenoma) detected on preoperative imaging, while BNE is performed in patients with MGD (>2 abnormal glands) or with negative imaging [17].

Several imaging studies are available for preoperative localization including ultrasound, ^99m^Technetium-sestamibi (MIBI), parathyroid four-dimensional computed tomography (4D-CT), parathyroid venous sampling, magnetic resonance imaging (MRI), and positron emission tomography (PET). While preoperative imaging is generally accepted as standard of care and recommended by society guidelines [17,18,19], there is a lack of consensus on the optimal diagnostic imaging algorithm, with variability in clinical practice. While most institutions use a combination of US and MIBI [18,20], an increasing number of centers have adopted 4D-CT and choline PET/CT in recent years. The choice of imaging exams ultimately depends on local radiologist expertise, surgeon preference, availability and costs [21].

A few general key points should be emphasized about preoperative imaging. First, diagnostic accuracy (i.e., sensitivity, specificity, etc.) for localizing abnormal glands can vary widely for the same type of imaging study. This is in part due to differences in imaging acquisition; for example, a “sestamibi” study can describe planar scintigraphic imaging alone or in combination with single photon emission computed tomography (SPECT), the latter improving diagnostic accuracy. Second, differences in patient population will result in differences in diagnostic performance. This is particularly important when comparing diagnostic accuracy between different modalities. All imaging studies will perform better in patients with SGD compared to patients with MGD [17,18], as solitary adenomas in SGD are significantly larger and more readily detected than hyperplastic glands in MGD. Cohorts with higher proportions of patients with SGD may in part explain the higher diagnostic performance reported in those studies and vice versa in cohorts with higher proportion of MGD. Therefore, a thorough evaluation of the patient population with respect to rates of SGD vs. MGD are important to factor in when comparing reported diagnostic parameters from different studies. Lastly, diagnostic performance can also vary depending on whether the method of analysis was performed per-patient or per-lesion/quadrant. Since patients have four parathyroid glands and any number of glands can be abnormal, imaging results and surgical reference standards have been analyzed using both methods in the literature, resulting in variability in diagnostic performance within and amongst imaging studies.

## 2. Ultrasound

Ultrasound (US) is one of the most commonly performed and established imaging techniques for preoperative localization in PHPT, as it is accessible, safe, accurate, and cost-effective. Furthermore, this technique is becoming ubiquitous in the practice of non-radiologists (such as endocrinologists and surgeons) who are increasingly competent with use of in-office ultrasound. US is widely accepted as the first-line imaging modality in the evaluation of patients with PHPT, often used in conjunction with MIBI (described below). The superficial location of the parathyroid glands in the central compartment of the neck permits the use of high frequency linear transducers, often with color doppler imaging to enhance detection of the abnormal parathyroid lesion.

Typically, the examination is performed with the patient’s neck hyperextended using both transverse and longitudinal imaging from the hyoid bone (cranially) to thoracic inlet (caudally) and extended to carotid arteries laterally [22]. The typical appearance of a parathyroid adenoma is a well-defined, ovoid, homogenous, hypoechoic (dark) lesion (relative to the thyroid tissue) in the usual location of a parathyroid gland (Figure 2). Adenomas are typically more vascular than thyroid nodules and lymph nodes. On Color Doppler, they may demonstrate a characteristic feeding vessel inserting on one of the poles and exhibit peripheral vascularity, as the feeding artery from the inferior thyroid artery enters from one pole and surrounds the periphery of the gland [23]. This feature is in contrast to a lymph node, which have an echogenic fatty hilum with a central blood supply, and can be used to differentiate between adenomas and nodes. Therefore, the use of Color Doppler increases the sensitivity of ultrasound [24]. Larger adenomas may be multi-lobulated, occasionally cystic or rarely calcified. Usually, normal parathyroid glands are not seen with US because of their small size.

The identification of intra-thyroidal parathyroid adenoma can be challenging, but is best performed with US. Characteristic ultrasound features are a solid, profoundly hypoechoic nodule (compared to other thyroid nodules or thyroid parenchyma) that may be partially or fully enveloped with thyroid tissue. As with other adenomas, a polar feeding vessel identified on Color Doppler within the nodule can facilitate the diagnosis. The operator should be more suspicious of an intra-thyroidal parathyroid adenoma when these characteristics are noted, especially when an extra-thyroidal parathyroid adenoma is not visualized [25].

The results of the diagnostic accuracy of ultrasound may be difficult to generalize, as the modality is operator dependent and will depend on individual operator expertise and experience. It is also difficult to determine given the heterogenous study population particularly if the studies included ectopic adenomas or had differing rates of MGD. A meta-analysis assessing the performance of ultrasound showed a pooled sensitivity or detection rate of 76.1% and positive predictive value (PPV) of 93.2% on a per-patient analysis. In this study, the definition of sensitivity was correct lateralization or correct identification of multi-gland disease. The definition of PPV was the probability of a positive test having a single adenoma or MGD. However, the analysis was limited by significant heterogeneity which diminishes the interpretability of these results, as evidenced by the wide range of reported sensitivity from 48.3% to 96.2% [26].

Evidence comparing the diagnostic accuracy of US compared to other modalities comes from paired test design. Most studies were observational, retrospective, and potentially limited by selection bias. For example, study population that consists of retrospectively selected patients who underwent PTx may overestimate the performance of the imaging modality, as a proportion of patients with negative scans may not to be referred for surgery [27]. Of preoperative localization techniques, US is most often compared to MIBI since both are well-established and traditional techniques. US and MIBI are generally considered comparable diagnostic methods to localize the abnormal PT gland [28]. In one observational cohort study [29], US and MIBI had similar performance with sensitivity of 76%, specificity of 97%, and negative predictive value (NPV) of 92% on a patient-based analysis, with PPV of 90% and 92% and overall diagnostic accuracy of 91% and 92% for US and MIBI, respectively.

While US is fairly accurate, it may not detect the lesion(s) in a significant proportion of patients and additional imaging is then required, most commonly MIBI. Conventionally, US and MIBI are complementary with US being an anatomical study, while MIBI provides for additional functional information and is more sensitive for ectopic adenomas, especially in the mediastinal location. When used in combination, both these techniques can improve accurate localization of the culprit parathyroid lesion. For example, MIBI localized the abnormal gland in 19 of 32 patients in which a lesion was not detect on US, recognized MGD in 2 of 5 patients that US did not identify as MGD and guided the surgeon to the correct side of the neck in 9 of 16 patients in whom US did not lateralize to the correct side. Using both US and MIBI increased the localization yield by 13% [29].

The main advantages of US include the absence of ionizing radiation, non-invasive technique and relative affordability compared to other imaging modalities. The US exam is relatively quick compared to the long image acquisition of a MIBI scan and its portability allows for bedside and in-office imaging. The disadvantages of US include that it is less sensitive at detecting MGD or ectopic glands [30]. In addition, US has relatively poor penetration of air and bone, limiting the detection of deeper glands around the trachea and esophagus in the neck and mediastinum. Ultrasound tends to be less accurate in patients with less severe disease (i.e., lower calcium and PTH levels), smaller adenomas, larger body habitus, or concomitant nodular thyroid disease [31].

## 3. Sestamibi Scintigraphy

While there is no current universally accepted algorithm for imaging localization, most institutions use a combination of MIBI scintigraphy and ultrasound [19,20]. Concordant localization on these two modalities is associated with a very high PPV [32] and widespread availability and cost-effectiveness make this strategy an attractive option for most centers [19].

MIBI scintigraphy is performed by intravenous injection of a radiotracer called ^99m^Technetium-Sestamibi (Tc-MIBI) and then imaging with a nuclear medicine gamma camera. Tc-MIBI is a lipophilic cation and localizes to hyperfunctioning parathyroid glands by accumulating within mitochondria-rich oxyphil cells found in parathyroid adenomas and hyperplastic glands [20,33,34]. Several different imaging protocols exist in clinical practice, including variations in timing (dual-phase MIBI imaging), additional radiotracer (dual-isotope subtraction), and type of acquisition, i.e., planar versus single photon emission computed tomography (SPECT). The different techniques and terminology can be quite confusing to the non-nuclear medicine physician and this heterogeneity also limits our ability to accurately determine the diagnostic performance of MIBI.

Dual-phase MIBI imaging is performed using two time points of imaging at early (10–30 min) and delayed phases (90–150 min) after MIBI intravenous injection [20]. In addition to parathyroid glands, MIBI also accumulates in the thyroid glands. However, there is differential washout of MIBI (Figure 3), with faster washout from the thyroid glands and delayed washout and retention within the parathyroid glands [35]. Dual-phase imaging takes advantage of the differential washout for imaging as both thyroid gland and abnormal parathyroid glands will be visualized on early phase images, however, only parathyroid glands will retain MIBI on delayed images, while the radiotracer will clear from the thyroid gland [33]. This technique is the most performed and will be the focus of the review.

Dual-isotope subtraction imaging is performed by injecting two different radiotracers, MIBI and either ^123^Iodine (^123^I) or ^99m^Technetium (^99m^Tc). Both ^123^I and ^99m^Tc only localize to the thyroid gland and therefore serve as a visual map of the thyroid gland. Two sets of images are acquired and then digitally subtracted from one another, with any persistent foci of uptake representing abnormal parathyroid glands. There is some literature that suggest dual-isotope subtraction imaging is superior to dual-phase MIBI [36]; however, the subtraction method requires two radiotracers with additional radiation dose to the patient [33].

Both dual-phase or dual-isotope subtraction methods can be imaged with planar or SPECT acquisition methods. Planar imaging results in two-dimensional images of the patient’s neck and chest, typically with an anterior view and possibly oblique and/or lateral views [33]. Planar imaging has largely been replaced with SPECT imaging, which provides axial tomographic three-dimensional images and can be reconstructed into coronal and sagittal views [35]. An analogy that may be helpful is that planar imaging is to a chest radiograph as SPECT imaging is to a CT chest. SPECT can be combined with CT images for fused SPECT/CT images for improved anatomic localization and have been shown to be superior to planar and SPECT alone [37].

Given heterogenous protocols, the diagnostic accuracy of sestamibi may be difficult to determine. A recent meta-analysis evaluating MIBI SPECT/CT, including studies using dual-phase or dual-isotope subtraction, showed that the pooled sensitivity or detection rate was 88% and 88% on a per-patient and per-lesion basis, respectively [35]. However, it is important to note that there were a limited number of patients with multigland disease in the studies included in this meta-analysis and almost all had solitary adenomas, which will skew the sensitivity of MIBI higher, as localization accuracy is worse for smaller hyperplastic glands in multigland disease compared to larger solitary adenomas. For example, in a study of 400 patients (80% single-gland disease and 20% multigland disease) by this review’s authors, MIBI SPECT/CT had an overall sensitivity of 58%, with a sensitivity of 75% in single-gland compared to 31% in multigland disease [22], which demonstrates how studies with lower or higher rates of multigland disease will have higher or lower reported diagnostic accuracy, respectively.

Advantages of MIBI include widespread availability of imaging acquisition and reader expertise and may show ectopic and deep/posterior glands that may be missed by ultrasound. MIBI can be used in patients who are unable to receive iodinated CT contrast and undergo 4D-CT due to a contrast allergy or renal failure [38] and has a lower radiation dose than 4D-CT [39]. Disadvantages include patient inconvenience given long imaging acquisition and wait times between early and delayed timepoints, more expensive than ultrasound or 4D-CT [40], atypical washout patterns hindering interpretation, and false-positive accumulation in thyroid nodules [20].

## 4. Parathyroid 4D-CT

Parathyroid 4D-CT is a relatively newer localization technique compared to MIBI and ultrasound and uses serial CT images to detect parathyroid adenomas and hyperplastic glands. First reported by Rodgers et al. in 2006 [41], 4D-CT was initially used an adjunct to MIBI and US but has since emerged as a first-line imaging modality for preoperative localization [42,43]. Studies have shown the excellent diagnostic accuracy in various clinical settings [42,43], including in the first-line setting, adjunct for inconclusive US and/or MIBI [44], and reoperative setting for persistent/recurrent PHPT [42,45], as well as in more difficult to localize populations such as patients with normocalcemic PHPT [46].

Despite its high-tech name, 4D-CT is simply a triple-phase CT of the neck and upper chest, with CT acquisitions obtained in noncontrast, arterial, and delayed phases, similar to that of triple-phase CTs of other body areas (i.e., CT of the liver). Regarding the four dimensions (“4D”), the first three dimensions refers to the three anatomic body planes of CT images (axial, coronal, and sagittal), with the fourth dimension referring to contrast enhancement over time on serial phases. Classically, a parathyroid adenoma or hyperplastic gland will demonstrate low density on noncontrast images compared to the thyroid, which is hyperdense due to high iodine content, intense enhancement on arterial phase, and rapid washout on delayed phase (Figure 4) [13]. Conversely, lymph nodes, a common mimicker of parathyroid glands, will show milder enhancement on arterial phase, which continues to slightly increase or plateau on delayed phase [13,20]. Interestingly, the aforementioned classic enhancement pattern of parathyroid adenomas is actually less common compared to other variant patterns [47], even though it is still the most commonly described pattern in the literature.

The conventional 4D-CT imaging acquisition protocol consists of CT images acquired from the angle of the mandible down to the carina, which includes all potential areas of eutopic and ectopic parathyroid glands [17]. Prior to scanning, proper patient positioning is key through hyperextension of the neck as tolerated to maximize exposure of the thyroid and parathyroid glands. This is most easily achieved by placing a small rolled sheet behind the top of upper back. Following acquisition of noncontrast images, 75 mL of iodinated CT contrast is injected intravenously (IV) at 4.0 mL/second, with an optional 25 mL normal saline chaser to flush any residual contrast in the subclavian vein to prevent streak artifacts. CT images are acquired at 25–30 and 60–80 s after the start of IV contrast injection for arterial and delayed (venous) phase images [13,22], with the time delay depending on individual CT scanners. Specific CT scan parameters (i.e., kVp) are well-described in the literature [13,20]. Axial images are obtained at 1-mm slice thickness, with 1.25–2.5-mm coronal and sagittal constructions [13,22]. Thus, contrary to popular belief, 4D-CT does not require specialized advanced equipment and can typically be performed at any imaging center with a 16-slice or higher CT scanner and IV contrast injector. Compared to MIBI, 4D-CT protocols have less variation and heterogeneity, allowing for more accurate comparison of diagnostic accuracy between different centers and published literature. The main variants are the addition of a fourth phase, the late delayed phase, acquired at 90–105 s minutes after IV contrast [48,49], or using only a single contrast-enhanced phase [50].

Interpretation of 4D-CT begins with reviewing the arterial phase in all three planes, as adenomas and hyperplastic glands are usually most conspicuous on this phase. Review should be targeted first towards eutopic anatomic locations of parathyroid glands near the thyroid gland and then expanded to ectopic locations (Figure 5), if not found [39]. Any suspected glands on the arterial phase are compared to noncontrast and delayed phase for serial pattern of enhancement over time. Progressively increasing enhancement, as opposed to rapid washout, and presence of a fatty hilum can be used to differentiate lymph node from parathyroid gland [39]. In addition, an enlarged vessel may be seen at the upper or lower poles of the parathyroid adenoma, which is a supporting finding called the “polar vessel” sign and found in about two-thirds of adenomas [51]. Some centers routinely report the size and weight of suspicious parathyroid glands and distances from anatomic landmarks (i.e., thyroid, esophagus, carotid artery, etc.) for the surgeon to use in preoperative planning [13,15].

As in other modalities, reported diagnostic accuracy of 4D-CT should be interpreted in light of the study patient population, biochemical profiles, and MGD rates. Two meta-analyses on preoperative imaging have included 4D-CT [26,52]. Cheung et al. performed a comprehensive meta-analysis comparing ultrasound, sestamibi SPECT and 4D-CT, with pooled sensitivity and PPV of 76.1% and 93.2%, 78.9% and 90.7%, and 89.4% and 93.5%, respectively. However, this study only included two studies on 4D-CT as it was still a relatively new technique at time of publication [26]. A more recent meta-analysis comparing 4D-CT to MIBI SPECT/CT showed sensitivity and specificity of 85% and 93% vs. 68% and 98%, respectively [52]. The pooled sensitivity of MIBI SPECT/CT is lower in this meta-analysis compared to the previously referenced one on MIBI SPECT/CT [35] (68% vs. 88%, respectively) [35,52], with the lower sensitivity most likely due to higher rates of MGD in the included studies and again highlighting the effect of patient population. However, this meta-analysis only includes studies that performed both 4D-CT and MIBI SPECT/CT and does not include all 4D-CT studies [52]. While there is still some debate, 4D-CT appears to have higher diagnostic localization accuracy than ultrasound and sestamibi, accounting for patient population.

The main advantage of 4D-CT is the high spatial resolution and excellent anatomic detail for precise localization and delineation of important anatomic landmarks for the surgeon. 4D-CT is performed very rapidly (~2 min) compared to MIBI (~3 to 4 h), which is more convenient and easier for patients. Unlike ultrasound, 4D-CT is not user dependent and allows for visualization of deeper and ectopic parathyroid glands [19]. Disadvantages include limited availability due to lack of radiologist expertise [53], atypical enhancements patterns [47], and lower accuracy in MGD and smaller adenomas, although relatively higher compared to other modalities [22]. Administration of IV contrast limits its application in patients with CT contrast allergy and renal failure. 4D-CT also has a higher radiation dose than MIBI, with effective doses of 10.4–28 millisieverts (mSv) and 7.8–12 mSv reported in one study [39,54]. However, is it important to remember that PHPT affects patients with mean age of 56–64 years old and thus are at lower risk from the effects of radiation exposure [39]; however, for younger patients, 4D-CT should be used judiciously [17].

4D-CT has been increasingly utilized in clinical practice as a first-line and adjunct imaging modality for preoperative localization. Recent society guidelines now propose 4D-CT as a potential option to consider in addition to ultrasound and MIBI [17,19].

## 5. Parathyroid Venous Sampling

Selective parathyroid venous sampling (PVS) is an invasive localization technique, which is now typically reserved for patients who have persistent or recurrent PHPT after PTX and/or who have had negative or discordant imaging studies. Usually performed by an experienced interventional radiologist who obtains various venous samplings from veins surrounding the thyroid bed and draining the anterior mediastinum. Sampling includes the internal jugular, brachiocephalic, internal mammary, vertebral, and thymic veins is and performed via peripheral vein catheterization. Sampling of the superior, middle and inferior thyroid veins may also be attempted to provide more precise localization. Laboratory support with use of rapid PTH assay is helpful to facilitate this procedure contemporaneously. Detailed knowledge of the anatomy of the typical and variant venous drainage of eutopic and ectopic parathyroid glands is paramount. The ultimate aim of the procedure is to regionalize the abnormal gland(s). An abnormal gradient is defined as ≥2-fold increase in PTH levels compared to the inferior vena cava [55] and the location of the hypersecreting gland inferred from the territory of the draining vein.

Correct lateralization (left or right) is deduced when abnormal PTH gradient is derived from each of the four quadrants of the neck. Subsequently, more precise localization can be mapped from where the abnormal gradient occurs around the thyroid bed. Superior thyroid veins drain into the upper internal jugular veins and an abnormal gradient here suggests the superior parathyroid gland as the source of excess PTH. Middle thyroid veins drain into the lower internal jugular veins, suggesting a superior or inferior parathyroid lesion. Inferior thyroid veins drain into the brachiocephalic vein, suggesting an inferior parathyroid lesion. An important variant to note is, paired inferior thyroid veins drain may into a common trunk before draining into the left brachiocephalic vein and thus, an abnormal PTH gradient here may not lateralize to the correct side. Another important caveat is that contralateral flow of blood may occur within the thyroid veins and inference about the lateralization may confounded by this. Abnormal PTH gradient from the vertebral vein suggests an inferior, intra-thyroidal, or retroesophageal parathyroid lesion. If venous sampling from thymic vein or internal mammary vein is abnormal, then a mediastinal or thymic parathyroid lesion is likely [56].

Given the variability of sampling protocols, venous drainages patterns, as well as indications for selective parathyroid venous sampling, its diagnostic accuracy may be difficult to determine. A recent meta-analysis of 12 pooled studies showed a pooled sensitivity of 74%, specificity of 41%, and positive likelihood ratio of 1.55 and for negative likelihood ratio was 0.47 [57]. However, the analysis was limited by significant heterogeneity which limits the interpretability of the results. The sensitivity of parathyroid venous sampling from the included studies range from 32% to 100%.

The advantages of selective parathyroid venous sampling are that it can guide surgery for patients who have failed a previous PTx or localize a culprit parathyroid gland if non-invasive imaging modalities have been negative or inconclusive [58]. This is especially important for patients with recurrent or persistent disease who desire a surgical cure, since repeat PTx is more difficult and can lead to a higher risk of complications and non-invasive techniques may be less accurate in such a scenario [59]. The disadvantages include its limited availability due to lack of radiologist expertise, invasive procedure, risk of intravenous iodinated contrast (similar to CT contrast), and risk of catheterization-related complications such as bleeding (i.e., groin hematoma) or thrombosis.

## 6. Other Newer Modalities

### 6.1. Choline PET/CT

Positron emission tomography (PET) is a non-invasive diagnostic imaging technique in nuclear medicine that can provide high quality images following intravenous injection of a radioactive tracer [60]. Unlike CT, which only provides anatomical detail, PET provides additional functional and metabolic information [61,62]. PET images can be combined and fused with CT images to improved anatomic localization as PET/CT, which is analogous to SPECT/CT used in MIBI scintigraphy. Different PET tracers are used for various imaging purposes, depending on the physiologic process being targeted within the body. The most commonly used PET tracer worldwide is 18F-Fluorodexyglucose (FDG); however, FDG PET has limited detection of benign parathyroid adenomas [63] but has some role in parathyroid carcinomas [64]. While several other PET tracers have been studied for parathyroid imaging, choline PET has the highest diagnostic performance and clinical utility and will be the focus of this review.

For PET imaging, choline can be radiolabeled with a positron emitter, typically 11-Carbon (^11^C-Choline) or 18-Fluorine (^18^F-Fluorocholine or FCH) [65], and used to image choline physiology within the body. Rapidly proliferating malignant cells have an increased demand for choline because of their requirement for enhanced phospholipid synthesis, a component of the cell membrane [66]. However, increased uptake of choline may also be seen in abnormal, but benign parathyroid cells due to upregulation of choline kinase related to the secretion of parathyroid hormone [66,67].

Overactive parathyroid glands were initially discovered incidentally in prostate cancer patients imaged with FCH PET/CT for the evaluation of their malignancy [68,69]. Since this discovery, dedicated FCH PET/CT for preoperative localization in PHPT has demonstrated promising results in localizing parathyroid lesions not detected on first-line imaging [70,71,72], with clinical utility in surgery-naïve patients undergoing primary PTx [72] and in patients undergoing reoperation after unsuccessful PTx [68,73].

Since FCH is more widely used than ^11^C-Choline, which requires an on-site cyclotron, we will discuss PET imaging using FCH. Imaging acquisition is typically performed 60 min after intravenous injection of FCH with the patient in a supine position on the PET/CT scanner. For interpretation, visual analysis of focal lesions detected with FCH-PET/CT is performed and maximum standardized uptake value (SUVmax) can be used for semiquantitative interpretation [73]. Focal uptake of radiotracer can be seen within a solitary (Figure 6) or multiple hyperfunctioning parathyroid glands in the thyroid bed or in an ectopic location (such as the mediastinum) [72]. The SUVmax of a parathyroid gland is usually greater than four times that of adjacent thyroid tissue [71].

A recent meta-analysis of 14 studies and 517 patients evaluated the diagnostic performance of FCH PET/CT. On a per-patient based analysis, overall sensitivity was 95% and PPV 97% while on a per-lesion analysis, pooled sensitivity and PPV were 92% and 92%, respectively [74]. FCH PET/CT may localize an abnormal parathyroid gland in 92% of patients with a negative US and/or MIBI imaging [72].

FCH PET/CT may have a role in detecting the culprit lesion(s) in patients with persistent or recurrent PHPT in which prior localization studies may have been equivocal or negative. For these potentially challenging patients, FCH PET/CT can provide added diagnostic value compared with conventional imaging [73]. For example, in patients with negative or discordant results on MIBI imaging and neck ultrasound, FCH PET/CT may detect true-positive uptake in lesions with a sensitivity of 99% and PPV of 91% [75].

There are several benefits that FCH PET/CT holds over traditional first-line imaging modalities. First, FCH PET/CT has a lower radiation dose than both MIBI SPECT/CT and 4D-CT, with an effective dose of approximately 6 mSv compared to 8 mSv and 12 mSv, respectively, while also demonstrating excellent spatial resolution facilitating the detection of small adenomas [59,76,77]. In addition, the scan duration is considered more patient-friendly due to its single acquisition scanning time of approximately 8 min [77] compared to the longer dual-time point imaging required for MIBI. Further, FCH PET/CT may result in less false positive results than US and MIBI as choline is taken up intensely in parathyroid adenomas and less in adjacent tissues, such as benign nodular thyroid disease, and therefore may reduce the risk of unnecessary neck explorations [71,77,78,79,80]. However, false positive intrathyroidal uptake may still occur in the context of thyroid malignancy [71,81,82]. A recent cost-effectiveness analysis on patients with non-localized PHPT showed that use of advanced imaging methods is more cost-effective than routine bilateral neck exploration [83].

Notwithstanding these significant strengths, FCH PET/CT also has some limitations. Currently, there is no accepted standardized protocol for imaging timing, dynamic acquisition phases, or the radiotracer dose [84,85]. FCH PET/CT has higher costs compared with conventional imaging modalities. However, costs can potentially be lowered if an on-site cyclotron is available and if a hospital functions as a referral center with frequent use of the technique. In addition, it is likely that the costs of PET/CT will further decrease, with newer PET cameras and software, requiring lower doses of PET radiotracers [84,86]. While there has been some suggestion that image guided MIP is more cost effective than BNE in patients with discordant first line imaging (US and MIBI) [87], this remains to be seen with FCH PET/CT. Although choline-radiolabeled agents are utilized for the imaging of prostate cancer patients, there remains an issue regarding availability of radiotracer and equipment in all imaging centers [84,88]. Lastly, although widely accepted for routine clinical use in European centers for parathyroid imaging, currently FCH PET/CT is not FDA-approved for this purpose in the United States and remains a research imaging technique [71].

No societal guidelines for the use of FCH PET/CT have been proposed as of yet, with its use largely determined by local availability and imaging expertise. Future directions for FCH PET/CT include further investigation of dual-time-point imaging. A recent study found that hyperfunctioning parathyroid glands were adequately visualized on early imaging in most patients, but a small subset (11%) were better seen at a later time point [86]. Two recent studies combined FCH PET and 4D-CT for integrated fusion imaging, with high sensitivity of 93–100%. In one study, combined FCH PET/4D-CT was shown to be more sensitive than either modality alone [89], while another study showed that combined FCH PET/4D-CT was more sensitive than 4D-CT alone but was equivalent to FCH PET alone [90]. FCH PET had a higher sensitivity than 4D-CT when directly compared [90].

### 6.2. MRI

Similar to PET/CT, magnetic resonance imaging (MRI) is a diagnostic imaging modality that provides both anatomical and functional information [91]. By taking advantage of randomly aligned spinning protons (abundant in water and fat), MRI can produce detailed images from any part of the body when a volume of tissue is subjected to a magnetic field resulting in the uniform alignment of protons, producing a magnetic vector [92]. When additional energy in the form of a radiowave frequency (RF) source is added to the magnetic field, the magnetic vector is deflected [93]. When this same source is switched off, the magnetic vector returns to its resting state and receiver coils around the body detect the emitted signal, which is then plotted on a grey scale producing cross sectional images [93,94].

Different tissues (such as fat and water) produce different signals based on the pulse sequences used (i.e., T1 and T2) and so can be identified separately. Many pathologies affect water content and cellular water motion, as well as angiogenesis and capillary permeability, which can be analyzed during image interpretation [93,94]. MRI has demonstrated some potential for detecting abnormal parathyroid glands, given its excellent soft tissue contrast, high spatial resolution and ability to evaluate the entire neck and mediastinum at the time of imaging [95].

MRI has been used as a second-line modality in preoperative parathyroid localization, in the case of negative or discordant first-line imaging modalities or in the context of persistent or recurrent PHPT, or a history of previous surgery. Although less commonly used for preoperative localization than US and MIBI, MRI has shown similar sensitivity in detecting abnormal parathyroid glands. In particular, MRI may be useful in imaging younger patients with parathyroid disease due to non-exposure to ionizing radiation. Dynamic MR imaging has also shown clinical utility in identifying parathyroid glands in the reoperative neck [96,97].

Imaging of the neck and parathyroid glands is typically performed with an anterior neck surface coil from the hyoid bone to the sternal notch using a 1.5 or 3 T magnet, the latter considered preferential due to increased image quality and water–fat separation techniques resulting in increased lesion detectability [95,98,99,100].

On MRI, benign parathyroid lesions typically demonstrate a well-defined border and an elongated morphology, usually best evaluated on T2 weighted imaging (Figure 7) [100]. In addition, a cleavage plane is commonly seen separating the parathyroid gland from the thyroid gland on the T2 out-of-phase sequence [100]. Parathyroid lesions can show variable signal characteristics on T1 and T2 weighted imaging but are usually bright (hyperintense) and homogenous appearing on T2 (96%). A marbled appearance can also sometimes be seen [100]. Following intravenous gadolinium contrast administration, rapid enhancement in the arterial phase is observed [101,102], which has been shown to increase diagnostic confidence [100,103]. Rarely, hemorrhage, fat (such as in a lipoadenoma) or fibrosis may result in atypical imaging features on MRI [98,100].

Diffusion-weighted imaging (DWI) is an MR imaging technique used in tumor imaging that measures the random motion of water molecules within a voxel of tissue. In general, highly cellular tissues or those with cellular swelling demonstrate bright signal with corresponding low signal on the corresponding apparent diffusion coefficient (ADC) map. Parathyroid lesions usually demonstrate higher (brighter) signal on DWI with increasing diffusion strength which can allow their differentiation from other soft tissues structures of the head and neck region [93,99,104].

The diagnostic accuracy of MRI for detecting parathyroid lesions varies with magnet field strength of the MRI scanner: For 1.5-Tesla scanners, MRI has a sensitivity of approximately 80% in most studies [105,106,107,108]. With 3-Tesla and using “4-D” (time resolved enhancement kinetics) techniques, the sensitivity of MRI increases to over 90%, greater than both ultrasound (76%) and MIBI (71%), [109], with the combination of all three modalities having a sensitivity of 100% [106,109]. Four-dimensional MRI had an overall 85% accuracy in distinguishing SGD from MGD [110], with an overall sensitivity of 74% for laterality and 77% for quadrant localization. In addition, 4D MRI has shown promise in accurately identifying “double adenomas”, a recognized risk factor for developing recurrent PHPT after PTx [110,111].

Eliminating the use of ionizing radiation is the main advantage of MRI. Compared to 4D-CT (which can be associated with radiation doses of up to 10 mSv) [59,110,112], this is a significant advantage as radiation exposure is the main risk for development of thyroid cancer, especially in younger patients, whose thyroid glands are more sensitive to the carcinogenic effect of ionizing radiation, and with thyroid doses of 50–100 mGy [113]. In addition, MRI provides excellent anatomic detail and may better identify small parathyroid gland lesions, as well as reduce the number of potential false positives found in the neck by better differentiating the parathyroid glands from adjacent lymph nodes and thyroid tissue [95,109,110,114]. Furthermore, MRI may be helpful in the differentiation of benign from malignant parathyroid lesions [10,104,114].

MR imaging of parathyroid glands also has some disadvantages. First, MRI often takes longer time to acquire images and is susceptible to artifacts from breathing and motion [101]. Efforts in recent years have been made to overcome these limitations by utilizing faster imaging tools and pulse sequences (TWIST) [115] and CAIPIRINHA [102,116]. Other disadvantages include the relative contraindication for MRI in select patients with implantable metal devices and the risk of nephrogenic systemic fibrosis and exacerbating kidney injury in patients with renal compromise following administration of gadolinium-based contrast agents [84,100]. Nonetheless, it has potential applications in cases of subsequent operation, difficult localization, or a contraindication to ionizing radiation [19].

The armamentarium of available imaging tests for PHPT has undergone significant advancement in recent years, including the adoption of PET/CT and MRI. The evolution of hybrid imaging techniques such as PET/MRI combining both these modalities to provide excellent anatomical resolution as well as physiologic information will likely further advance this field [117]. FCH PET/MRI is a novel hybrid imaging technique that has only recently been evaluated in pilot study of 10 patients with PHPT and nonlocalized disease with negative or inconclusive results on conventional imaging. FCH PET/MRI was found to localize abnormal PT glands with a 90% sensitivity and 100% PPV [118]. Larger studies should be performed to determine the diagnostic accuracy of FCH PET/MR for localization of abnormal parathyroid glands.

## 7. Conclusions

Recent advances in preoperative localization of abnormal parathyroid glands in PHPT have allowed more accurate and potentially curative minimally invasive surgical treatments for patients. With these advancements, endocrinologists and surgeons treating patients with PHPT now have a battery of imaging tests and techniques to choose from. As such, physicians should familiarize themselves with this array of techniques to provide the most suitable imaging studies to optimize patient care, while also considering local expertise and experience with these different modalities.

## Figures and Tables

**Figure 1 biomedicines-09-00390-f001:**
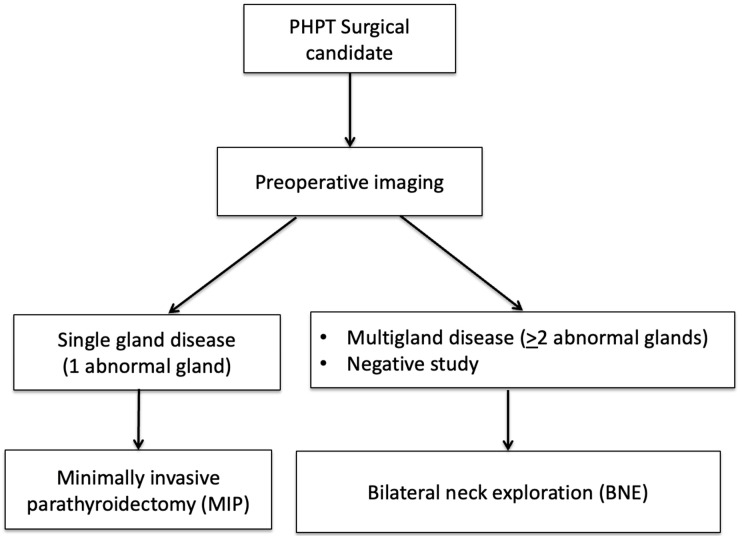
How imaging is used to guide parathyroid surgery.

**Figure 2 biomedicines-09-00390-f002:**
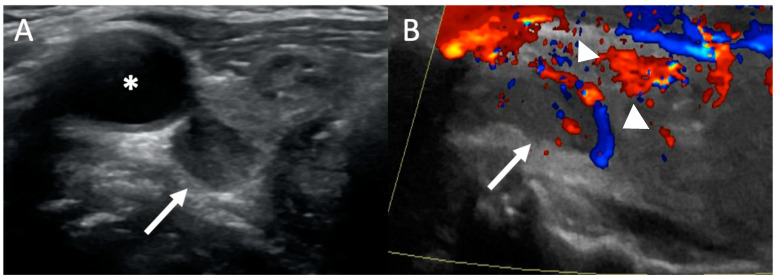
Ultrasound: 54-year-old woman with primary hyperparathyroidism (PHPT). Transverse grayscale (**A**) ultrasound image demonstrates an ovoid hypoechoic lesion (white arrow) posterior to the right lobe of the thyroid gland and carotid artery (*) consistent with a parathyroid adenoma. (**B**) Color doppler ultrasound image shows a polar feeding vessel with peripheral vascularity (white arrowheads).

**Figure 3 biomedicines-09-00390-f003:**
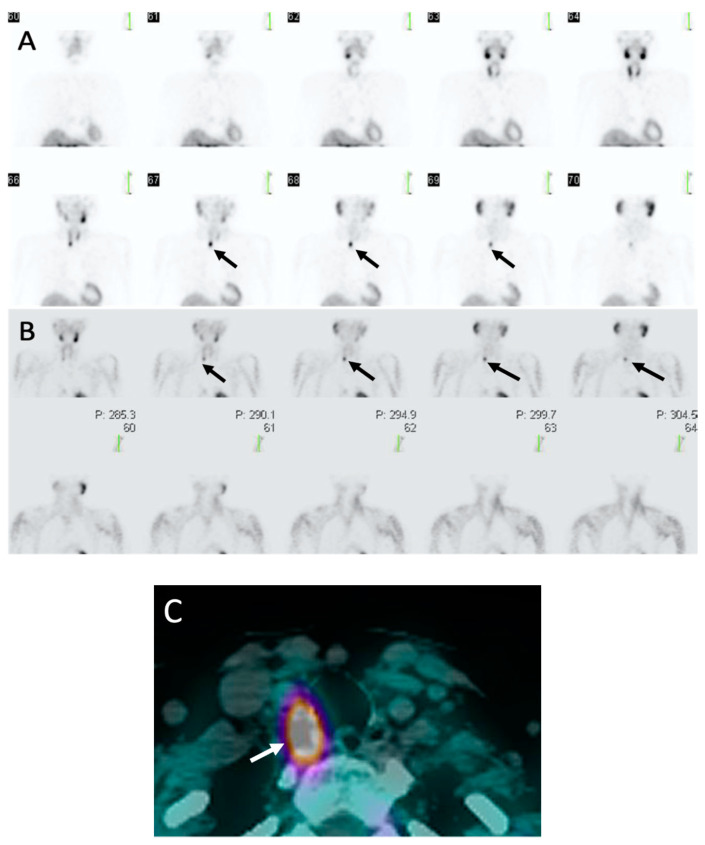
Sestamibi: 62-year-old woman with PHPT. (**A**) Coronal early phase sestamibi single photon emission computed tomography (SPECT) images demonstrate uptake in both the thyroid and parathyroid glands with asymmetric contour along the posterior right thyroid gland (black arrows).(**B**) Coronal delayed phase sestamibi SPECT images demonstrate radiotracer washout from the thyroid gland with retention of radiotracer posterior to the right thyroid gland consistent with a right upper parathyroid adenoma (black arrows) (**C**) Axial delayed phase sestamibi SPECT/computed tomography (CT) shows ^99m^Technetium-sestamibi (MIBI) uptake posterior to right thyroid gland (white arrow) corresponding with a right upper parathyroid adenoma. Addition of CT to SPECT allows for improved anatomic localization.

**Figure 4 biomedicines-09-00390-f004:**
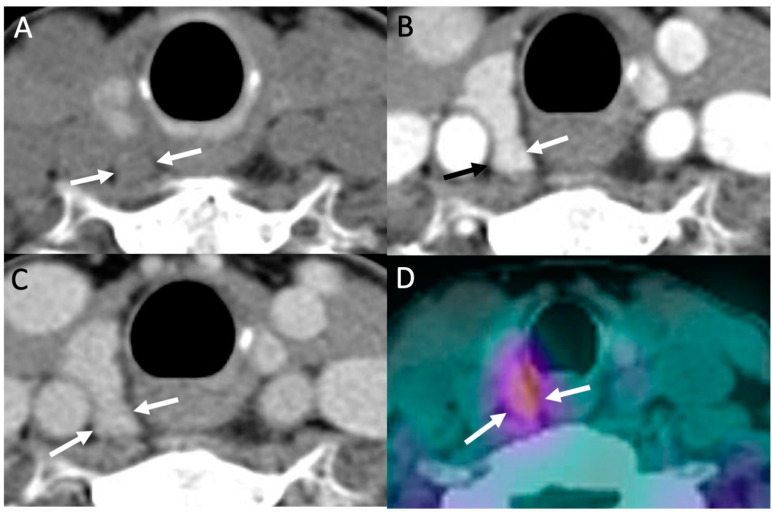
Parathyroid 4D-CT: 49-year-old female with PHPT. Parathyroid 4D-CT images in noncontrast (**A**), arterial (**B**) and delayed (**C**) phases demonstrating a low-density nodule (arrows) posterior to the mid right thyroid gland with avid contrast enhancement on arterial phase and washout on delayed phase, consistent with right upper parathyroid adenoma. (**D**) Sestamibi SPECT/CT performed the same day shows corresponding focal radiotracer uptake in the adenoma (arrows).

**Figure 5 biomedicines-09-00390-f005:**
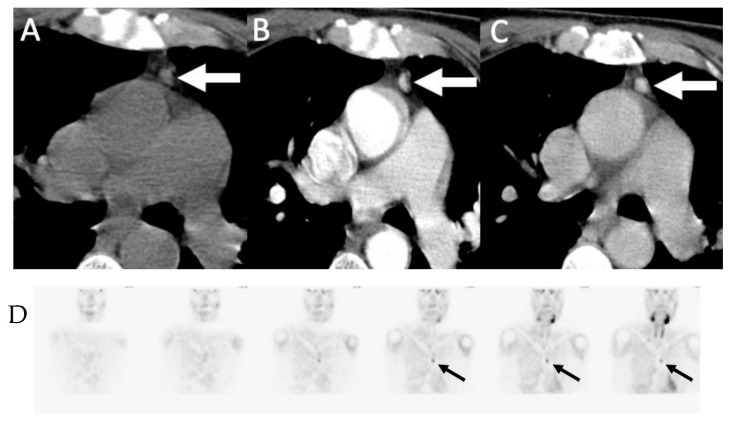
Ectopic mediastinal adenoma: 62-year-old man with primary hyperparathyroidism 4D-CT demonstrating a small enhancing nodule in the anterior mediastinum (arrows) seen on non-contrast (**A**), arterial-phase (**B**) and delayed phase (**C**) imaging without washout. (**D**) Coronal early phase sestamibi SPECT images demonstrate mild uptake in the anterior mediastinum, corresponding to the enhancing nodule on 4D-CT. The patient was subsequently taken for thoracoscopic resection which demonstrated an ectopic parathyroid adenoma within thymic tissue.

**Figure 6 biomedicines-09-00390-f006:**
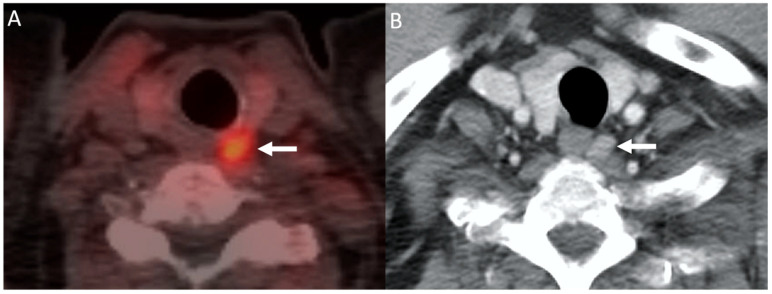
Choline PET/CT: 68-year-old man with hypercalcemia and history of prostate cancer. ^11^C-Choline-PET/CT image (**A**) demonstrating focal choline uptake within a nodule posterior to the left mid thyroid gland (arrow). Subsequent contrast-enhanced CT (**B**) demonstrating a corresponding enhancing nodule. The patient was subsequently taken for surgical resection which revealed a left upper parathyroid adenoma.

**Figure 7 biomedicines-09-00390-f007:**
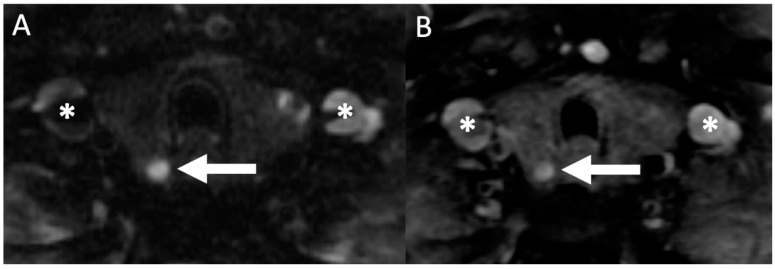
Magnetic resonance imaging (MRI): 71-year-old woman with hypercalcemia. Axial T2 weighted imaging (**A**) showing a hyperintense (bright) nodule posterior to the right thyroid lobe (white arrow) with subsequent axial (**B**) gadolinium contrast enhanced T1 weighted imaging showing an enhancing lesion (white arrow), subsequently resected confirming the presence of parathyroid adenoma on histopathology. Asterisks (*) denote the location of the bilateral jugular veins.

## Data Availability

Not applicable.

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
