# Peer review of "Preoperative Localization for Primary Hyperparathyroidism: A Clinical Review"

_biomedicines, 2021, doi:10.3390/biomedicines9040390_

Round 1

Reviewer 1 Report

Paper well written. I suggest to be less verbose in the description of every single technique highlighting better the role of new imaging modalities as PET/CT and 4DceCT and fusion imaging PET/CeCT (4D).

Minor revision in references considering the latest works on imaging fusion PET/CECT are required

Reviewer 2 Report

The article describes an interesting subject. It is a well - written review with the most recent and complete compilation of knowledge. It presents the subject in a smooth, interesting, logical and systematic way.

I do not find any major drawbacks in this review.
However it would be worth of mention that in some countries indium 111In is used instead of sestamibi and I didnt find in the article any information of the use of FDG-PET and nevertheless this is the tracer used in 92% of PET scans worldwide.
